# Extracellular Vesicles as an Index for Endothelial Injury and Cardiac Dysfunction in a Rodent Model of GDM

**DOI:** 10.3390/ijms23094970

**Published:** 2022-04-29

**Authors:** Stephanie M. Kereliuk, Fengxia Xiao, Dylan Burger, Vernon W. Dolinsky

**Affiliations:** 1Diabetes Research Envisioned and Accomplished in Manitoba (DREAM), Research Theme of the Children’s Hospital Research Institute of Manitoba, 715 McDermot Avenue, Winnipeg, MB R3E 3P4, Canada; umkereli@myumanitoba.ca; 2Department of Pharmacology and Therapeutics, University of Manitoba, Winnipeg, MB R3E 0T6, Canada; 3Kidney Research Centre, Ottawa Hospital Research Institute, Ottawa, ON K1H 7N9, Canada; fxiao@ohri.ca; 4Department of Cellular and Molecular Medicine, University of Ottawa, Ottawa, ON K1H 7N9, Canada

**Keywords:** extracellular vesicles, pregnancy, diabetes, gestational diabetes, mitochondria

## Abstract

Gestational diabetes mellitus (GDM) increases risk of adverse pregnancy outcomes and maternal cardiovascular complications. It is widely believed that maternal endothelial dysfunction is a critical determinant of these risks, however, connections to maternal cardiac dysfunction and mechanisms of pathogenesis are unclear. Circulating extracellular vesicles (EVs) are emerging biomarkers that may provide insights into the pathogenesis of GDM. We examined the impact of GDM on maternal cardiac and vascular health in a rat model of diet-induced obesity-associated GDM. We observed a >3-fold increase in circulating levels of endothelial EVs (*p* < 0.01) and von Willebrand factor (*p* < 0.001) in GDM rats. A significant increase in mitochondrial DNA (mtDNA) within circulating extracellular vesicles was also observed suggesting possible mitochondrial dysfunction in the vasculature. This was supported by nicotinamide adenine dinucleotide deficiency in aortas of GDM mice. GDM was also associated with cardiac remodeling (increased LV mass) and a marked impairment in maternal diastolic function (increased isovolumetric relaxation time [IVRT], *p* < 0.01). Finally, we observed a strong positive correlation between endothelial EV levels and IVRT (r = 0.57, *p* < 0.05). In summary, we observed maternal vascular and cardiac dysfunction in rodent GDM accompanied by increased circulating endothelial EVs and EV-associated mitochondrial DNA. Our study highlights a novel method for assessment of vascular injury in GDM and highlights vascular mitochondrial injury as a possible therapeutic target.

## 1. Introduction

Mammalian pregnancy is marked by distinct phases of metabolic and cardiovascular adaptations that facilitate fetal growth and development. As pregnancy progresses, plasma volume expands, peripheral resistance decreases and uteroplacental blood flow increases [1]. At the same time, insulin resistance increases in maternal tissues and maternal metabolism shifts towards the preferential utilization of fatty acids, in order to preserve glucose for the fetus [2]. After delivery, these maternal systems typically return to the pre-pregnant state.

In the presence of maternal obesity or excess gestational weight gain, the natural insulin resistance of pregnancy can become maladaptive and lead to hyperglycemia and glucose intolerance, that is characteristic of gestational diabetes (GDM). GDM is broadly defined as glucose intolerance with first onset during pregnancy and is one of the most common medical complications of pregnancy (up to 20% of all pregnancies worldwide) [3]. GDM is associated with risk of hypertension/preeclampsia, caesarean delivery, and macrosomia, as well as neonatal hypoglycemia and respiratory distress [4,5]. In most cases, blood glucose returns to pre-pregnancy levels, although women who have experienced GDM have a >7-fold increased risk of developing type 2 diabetes later in life [6,7]. The prevailing opinion has been that during the period of pregnancy, GDM is not associated with significant target organ damage or vascular alterations that are typically seen with longstanding diabetes. However, recent work by Shah and colleagues suggest that risk of microvascular injury may be higher in women with GDM, despite the relatively brief period of glucose intolerance [8].

Endothelial injury is one of the earliest consequences of diabetes and a common precursor to cardiac injury/dysfunction. Diabetes/hyperglycemia-associated related changes to the endothelium include increased reactive oxygen species (ROS) production, impaired nitric oxide (NO) release and a shift to a pro-inflammatory phenotype [9]. Arterial stiffening and impairment in endothelium-mediated vasorelaxation may also arise and progress to diastolic heart failure [9]. A key mediator of endothelial injury is mitochondrial dysfunction. While ATP production in endothelial cells is achieved through glycolysis, mitochondria play a central role in intracellular signaling, calcium homeostasis, amino acid synthesis, and meeting energy demands during proliferation or stress [10]. Mitochondria are also the primary source of oxidative stress in endothelial cells through superoxide anion generation during oxidative phosphorylation [11,12,13]. Under stress conditions, such as diabetes, mitochondrial ROS production is increased leading to impaired function, inflammation, and apoptosis [11,12,13]. Such effects are counterbalanced by activation of pathways that mediate resistance to oxidative stress such as nicotinamide adenine dinucleotide (NAD+)-dependent sirtuin deacetylases [14,15,16]. An additional mechanism by which mitochondria can disrupt endothelial function is by mitochondrial DNA (mtDNA) release and inflammatory responses [17]. To date, insights into maternal endothelial dysfunction in GDM have been limited and their link to cardiac function has not been explored extensively.

One emerging tool for the study of endothelial injury/dysfunction is the assessment of extracellular vesicles (EVs). EVs are membranous vesicles formed ubiquitously under physiological conditions as well as stress, injury, and death [18,19,20]. Once formed, EVs contain functional membrane and cytoplasmic proteins, lipids, nucleic acids and other molecules derived from their cell of origin [21]. EVs of various origins are found in biological fluids and their levels are altered under pathological conditions [22]. In particular, increases in levels of 100–1000 nm EVs from endothelial cells (often referred to as endothelial microparticles) have been reported in hypertension [23,24], chronic kidney disease [25], as well as type 1 and type 2 diabetes [26]. Levels of endothelial EVs are positively correlated with blood pressure and measures of endothelial dysfunction [23,25] and predictive of adverse cardiovascular events [27,28]. Recently, we reported that high levels of endothelial EVs were predictive of adverse pregnancy outcomes in women with type 1 diabetes [29]. Thus, EVs offer a novel approach for the study of endothelial injury/dysfunction in a non-invasive fashion.

EVs have also been studied in the context of GDM where much of the focus has been on placental-derived EVs and their potential role in insulin resistance [30,31,32,33,34]. In this regard, placental-derived EVs are increased in women who develop GDM [35,36] with alterations to miRNA [37] and protein composition [38,39]. With respect to circulating EVs, Franzago et al. previously reported an increase in the relative abundance of adipocyte-derived EVs [40]. GDM also appears to be associated with altered miRNA and protein composition in circulating EVs that may provide insights into disease pathogenesis [33,41,42]. However, to the best of our knowledge, the relationship between circulating EVs and the maternal molecular and functional alterations associated with GDM has not been explored.

In this study, we used a rat model of diet-induced GDM that resembles several aspects of a GDM pregnancy in humans including excess gestational weight gain, glucose intolerance and beta-cell dysfunction [43,44]. We assessed the impact of GDM on the maternal vasculature by quantifying circulating markers of vascular injury (i.e., endothelial EV levels, and EV-associated mitochondrial DNA) and determined the impact of GDM on maternal heart function. We observed increases in markers of endothelial injury/dysfunction coupled with cardiac remodeling and LV diastolic dysfunction. At the molecular level, we noted vascular NAD+ depletion which may contribute to functional impairment. Taken together these results suggest maternal vascular and endothelial dysfunction during this high-risk period and provide novel insights into possible pathogenic mechanisms.

## 2. Results

### 2.1. Maternal Characteristics

Prior to pregnancy, the lean and GDM females were similar weight, and pregnancy markedly increased the body weights of both groups of females (Figure 1A). By e18.5 of pregnancy, the body weight of the diet-induced GDM group was associated with a 10% greater body weight than lean females (Figure 1A, *p* < 0.05). This was a consequence of a 31% greater gestational weight gain in the GDM group compared to the lean females (Figure 1B, *p* < 0.01). In pregnancy, hyperglycemia and glucose intolerance were apparent in e17.5 GDM rats compared to lean dams (Figure 1C,D).

### 2.2. Measures of Vascular Injury

We next assessed levels of circulating EVs in lean and GDM females. Nanoparticle tracking analysis (NTA) showed that levels of total particles were not significantly different and there was no difference in the mean size of particles between lean and GDM rats (Figure 2A,B). Using flow cytometry, we further confirmed that there was no significant difference in total Annexin V positive EVs (Figure 2C). However, as shown in Figure 2D, GDM rats exhibited an ~3-fold increase in endothelial EVs suggesting increased endothelial cell stress. Consistent with this we also observed an ~4-fold increase in plasma von Willebrand factor (vWF) levels (Figure 2E).

### 2.3. Aortic NAD+, and Mitochondrial DNA Shedding

To explore possible mechanisms of endothelial injury, we examined the impact of GDM on vascular NAD+ levels, and EV-associated mtDNA as an indirect measure of mitochondrial damage. We observed a 53% reduction in NAD+/NADH ratio in aortas from GDM rats compared with their lean counterparts (Figure 3A). These changes were associated with an increase in EV-associated mtDNA suggesting ongoing mitochondrial stress (Figure 3B,C).

### 2.4. Maternal Cardiac Structure and Function

Finally, we used transthoracic echocardiography to examine the cardiac morphometry and function of the females prior to pregnancy and near the end of pregnancy on e18.5. As shown in Table 1, cardiac structures in the female rats were similar prior to pregnancy. At e18.5, pregnancy altered cardiac geometry, although several of these alterations in GDM rats were more pronounced. For example, left ventricular (LV) Mass in the GDM dams was ~14% higher than the lean dams (Table 1) which, when compared to lean dams, was associated with a 2.7-fold greater increase in LV Mass during pregnancy in the GDM group (Figure 4A, *p* < 0.05). The increase in LV mass was proportionate with gestational weight gain, as LV mass per gram of body weight was not different between the groups (Table 1). The increased LV mass was primarily due to significant increases in LV posterior wall and LV anterior wall thickness and not the intraventricular septal wall thickness in GDM dams, when compared to the lean dams (Table 1). GDM induced a 4-fold greater increase in LV anterior wall thickness (Figure 4B, *p* < 0.05) and a 3.2-fold greater increase in LV posterior wall thickness (Figure 4C, *p* < 0.001). We also examined LV systolic and diastolic function of the heart. Prior to pregnancy, both the systolic and diastolic function were similar in both groups of female rats (Table 1). At e18.5 of pregnancy, systolic functional parameters (e.g., ejection fraction, fractional shortening) in GDM and lean dams were the same (Table 1). On the other hand, evidence of LV diastolic dysfunction was apparent. GDM reduced the E’/A’ ratio and also markedly increased the isovolumetric relaxation time (IVRT, Table 1). The pregnancy-induced increase in IVRT was 2.7-fold higher in the GDM dams compared to the lean females (Figure 4D, *p* < 0.001). Moreover, the myocardial performance index was higher in GDM dams (Table 1), indicating the presence of cardiac dysfunction. Finally, we observed a strong positive correlation between endothelial EV levels and IVRT (Figure 4E, r = 0.58, *p* < 0.05) suggesting a connection between endothelial and cardiac dysfunction in our model.

## 3. Discussion

The present study assessed the impact of GDM on maternal vascular and cardiac health using a combination of circulating biomarkers, molecular characterization, and physiologic measures. The major findings are that rats with GDM exhibit evidence of endothelial injury and cardiac dysfunction. With respect to the vasculature, we observed increases in circulating endothelial EVs and plasma vWF suggesting endothelial stress. This was further evidenced by deficiency in NAD+ levels in aortas and increased levels of EV-associated mtDNA suggesting that mitochondrial injury may play a causal role in GDM-associated vascular injury. From a functional standpoint, we observed evidence of cardiac remodeling and diastolic dysfunction that was positively correlated with endothelial EV levels.

Maternal cardiovascular health is a critical determinant of fetal development and impaired vascular function is associated with adverse pregnancy outcomes and may contribute to future cardiovascular disease [5,45,46]. There is emerging evidence associating GDM with maternal endothelial [8,47] and cardiac [48,49] dysfunction, however approaches for identifying and quantifying injury are lacking and molecular insights in the context of GDM have been limited. Using a well-established rodent model of diet-induced obesity and GDM, we observed evidence of late pregnancy endothelial and cardiac injury. In particular, we observed a significant increase in circulating endothelial EVs. Endothelial EVs are an emerging biomarker of endothelial injury that strongly correlate with functional measures [23,50] and predict future cardiovascular morbidity and mortality [27,28]. Evidence of endothelial injury in our study is in line with work by Shah et al. that identified microvascular injury in human GDM [8] and is further supported by our observation of increased circulating levels of vWF, a pro-thrombotic protein released from stressed endothelial cells [51] in the GDM rats.

EVs are emerging as a novel tool for the study of GDM-associated pathological changes [30]. Indeed, changes to the formation or molecular composition of EVs from plasma [40,42,52], urine [53,54], and even oral (gingival crevicular) fluids [55] have been reported. Our study focused on plasma-derived EVs where we observed increases in circulating endothelial EV levels with no change in total EVs using nanoscale flow cytometry. Similarly, Franzago and colleagues assessed circulating EVs in human GDM by flow cytometry [40]. While they did not observe statistically significant increases in endothelial EV levels, there was a trend towards increased levels in women with GDM compared with healthy controls (*p* = 0.09). Our data, in a more controlled setting of experimental GDM supports an increase in endothelial EVs. Such an increase would be notable as we have recently shown that high levels of endothelial EVs predict adverse pregnancy outcomes in type 1 diabetes [29]. Similar to our observations, Franzago did not see changes in circulating total EVs, however, Salomon (2016) and Arias (2019) have previously reported increases in circulating total EVs in GDM [35,56]. The latter studies employed approaches that focused on small EVs specifically and it is possible that GDM may differentially impact on the various EV subpopulations. Alternatively, our study and that of Franzago may simply be underpowered to detect changes in total EVs as both studies showed trends towards increases in GDM.

In addition to increases in levels of circulating endothelial EVs, we also observed increases in circulating EV-associated mitochondrial DNA. To the best of our knowledge this represents the first study to examine mtDNA in circulating EVs in GDM. However, Jayabalan et al. recently examined circulating EVs in GDM by quantitative proteomics and noted enrichment in proteins associated with energy production [42]. Of particular note was the enrichment of calcium/calmodulin-dependent protein kinase II beta, which has been implicated in mitochondrial fragmentation in hyperglycemia [57]. Thus, alterations in EV composition may reflect mitochondrial injury in GDM. EVs thus offer an attractive non-invasive approach to monitoring mitochondrial injury. Nevertheless, the optimal approach is unclear and it is not known whether this may be used to monitor response to treatment, as has been reported for certain EV-associated miRNA [52].

Vascular health is tightly linked to cardiac function and endothelial dysfunction often precedes and predicts functional impairment of the heart [58,59]. This relationship is commonly observed in the context of chronic conditions such as obesity, hypertension, and type 1 and 2 diabetes [60]. However, what has been less clear is whether the brief period of glucose intolerance seen in GDM is associated with maternal cardiac dysfunction. Recently, Meera et al. employed speckle tracking echocardiography to assess cardiac function in GDM [48]. Using this approach, which allows for tracking of subtle alterations to myocardial function, they observed impairment in LV global longitudinal strain despite no changes in traditional measures of cardiac function. Buddeberg and colleagues also observed reduced longitudinal strain as well as ventricular remodeling in a cohort of women with GDM [49]. Thus, it appears that GDM-associated cardiac impairment may have been historically underappreciated with subtle changes not observed by conventional measures. Our results are consistent with the above studies. We also observed structural changes (i.e., LV mass, LVAW, LVPW) as well as an increase in IVRT in GDM rats compared with lean controls. Despite these changes, established indices of systolic cardiac function such as ejection fraction were not significantly different between groups. This observation is notable as heart failure with preserved ejection fraction (HFpEF) is a condition with increasing prevalence (particularly in women), a high mortality rate, and a lack of therapeutic options [61]. It is possible that the subtle changes to heart structure and function observed in our study contribute to future susceptibility to HFpEF, however, this remains to be studied. We also observed a strong positive correlation between IVRT and endothelial EV levels which may suggest an interrelationship between endothelial injury and subtle impairment in cardiac function associated with GDM. It is possible that a causal relationship exists whereby endothelial injury and EV release drive cardiac dysfunction. Conventionally, endothelial dysfunction precedes functional cardiac impairment and drives pathogenesis via coronary artery disease and altered vascular resistance [62]. It is possible that EVs accelerate this process through direct impairment of cardiac function [63]. Conversely, endothelial injury could be secondary to cardiac impairment in GDM or unrelated entirely. The nature of this relationship and whether these changes play a role in the short- and long-term risks associated with GDM is a subject for future investigation.

We also explored possible mechanisms for GDM-induced vascular injury. One emerging mechanism of endothelial injury is impairment in NAD+-mediated control of mitochondrial function [64,65,66]. We observed a significant decrease in NAD+ bioavailability in aortas from GDM rats as compared with their lean counterparts. While a previous report suggested reductions in NAD+ in fetal endothelial progenitor cells from GDM pregnancies [67], to the best of our knowledge, this represents the first report of NAD+ deficiency in the maternal vasculature in GDM. These alterations appear to impact on vascular mitochondrial health as we observed a significant increase in release of EV-associated mtDNA.

Several circulating biomarkers have been identified that may allow for monitoring of cardiovascular risk status and personalization of diabetes management [68]. Indeed, alterations in inflammatory mediators, circulating adhesion molecules, markers of oxidative stress, and lipoproteins have all been reported in GDM [68,69]. Our results suggest that circulating EVs may also serve as markers of vascular and cardiac injury in GDM. We observed not only increases in circulating endothelial EV levels, but also a change in their molecular composition with increased mtDNA levels. Circulating EVs may be particularly appealing as biomarkers because changes in their composition reflect those in their cell of origin [19]. However, it will be critical to confirm our observations in human GDM and determine whether circulating EVs offer value over other biomarkers of cardiovascular injury in GDM.

Our study has some limitations. There was a relatively small sample size. In addition, our rodent model of GDM was achieved through diet-induced obesity and, as such, the findings may not be generalizable to GDM in lean individuals. There are differences between rat and human pregnancy, including a larger number of offspring per pregnancy in rats, a shorter gestational length and, though increases in blood volume are characteristic to both, changes in hemoglobin are more pronounced in the rat [70]. In addition, placentas are structurally different in humans (one trophoblastic layer) and rats (three trophoblastic layers). At the biochemical level, while there are changes in circulating glucose, lipids, albumin, and electrolytes over the course of pregnancy, there are considerable similarities in the timing and magnitude of these changes in humans and rats. Nonetheless, future studies will need to examine EV levels and content in human GDM cohorts. We also note that our work is associative and does not establish a causal relationship between altered NAD+/mitochondrial signaling and the endothelial/cardiac dysfunction seen in this study. Nevertheless, our study also has a number of strengths. First, we employed a well-established rat model of GDM that enabled the comprehensive study of vascular and cardiac effects in a controlled environment. This approach may have allowed us to detect changes that would not otherwise be seen in human GDM (i.e., endothelial EV increases, cardiac alterations). We also employed a combination of emerging non-invasive measures of vascular health and sensitive functional measures of cardiac function, which may be useful in future interventional studies aimed at reducing the burden of GDM. Finally, we note that the focus on both vascular and cardiac effects is unique and allows for consideration of the interrelationship between these two variables that is not possible in studies where these systems are evaluated individually.

## 4. Materials and Methods

### 4.1. Maternal GDM Model

All procedures were approved by the University of Manitoba’s Central Animal Care Animal Welfare Committee in adherence to the principles for biomedical research involving animals developed by the Canadian Council on Animal Care and the Council for International Organizations of Medical Sciences. Rats were given ad libitum access to food and water, and housed two per cage.

A rodent model of diet induced obesity was used as previously described to induce gestational diabetes mellitus (GDM) in pregnancy [43,44]. Three-week-old female Sprague–Dawley rats were obtained from Central Animal Care Services of the University of Manitoba. Pre-pregnancy obesity was induced by feeding a high fat and sucrose (HFS) diet (45% kcal fat, 35% kcal carbohydrate with 17% kcal from sucrose; D12451, Research Diets, Inc., New Brunswick, NJ, USA) for six weeks prior to mating, while lean control females consumed a low fat (LF) diet (10% kcal fat, 70% kcal carbohydrate with 35% kcal from sucrose; D12450B, Research Diets, Inc., New Brunswick, NJ, USA) After six weeks of diets, LF- and HFS-fed females (nine weeks of age) underwent pre-breeding glucose tolerance tests and serum was collected and stored at −80 °C for later analysis. Female rats also underwent in vivo echocardiography prior to breeding to assess heart structure and function. Females were subsequently mated with chow-fed male Sprague–Dawley rats (seven to eight weeks of age, one male per two females). Pregnancy was confirmed 10 days after separation of male and female breeders by an abdominal ultrasound, and pregnant females were singly housed until birth. Therefore, two experimental groups of pregnant rats were generated: LF diet fed “lean” females and HFS diet fed “GDM” females, with diets maintained throughout pregnancy and lactation. Random and fasted blood glucose levels were measured from the tail vein, and serum was collected in each trimester and stored at −80 °C for later analysis. Maternal glucose tolerance tests were performed at e17.5. Fetal and maternal in vivo echocardiography was also performed at e18.5 to assess heart structure and function. Food consumption and body weight were measured weekly.

### 4.2. Metabolic Assessment in Pregnancy

Pregnant non-fasted dams were sacrificed at the end of gestation (embryonic day 20). Euthanasia was initiated with an intraperitoneal (IP) sodium pentobarbital (2 mL/4.5 kg) overdose injection and terminated by cardiac puncture with blood collection.

Blood was collected from pregnant dams by cardiac puncture using a 10 mL syringe and 18-gauge needle, transferred to BD Vacutainer^®^ tubes coated with 10.8 mg K2EDTA (BD, Franklin Lakes, NJ, USA) and spun at 10,000 g for 20 min at 4 °C. Serum was aliquoted, flash-frozen in liquid nitrogen and stored at −80 °C for future analysis. Tissues from pregnant dams were extracted, weighed, flash frozen in liquid nitrogen, and subsequently stored at −80 °C for future analysis.

Glucose tolerance tests (GTTs) administered by IP injection were performed as described previously with modifications [43]. Briefly, fasted blood glucose was measured from the tail vein at time 0 using an ACCU-CHEK^®^ Aviva glucose meter (Roche Diagnostics, Laval, QC, Canada). Blood was also collected from the tail vein at time 0 using a Microvette^®^ CB 300 Serum tube (Sarstedt, Nümbrecht, Germany). Glucose was then administered IP at a concentration of 2 g/kg body weight and blood glucose was subsequently measured at 15, 30, 45, 60, 90, and 120 min post injection. The concentration of blood glucose over time was plotted and the area under the curve (AUC) was calculated. IP GTTs were performed on non-pregnant females after being on their respective diets for six weeks (pre-breeding GTT) and during pregnancy at e17.5. Rats were fasted overnight for the pre-breeding GTT and for 4 h for the third trimester GTT. The fasting period was shortened to 4 h in the third trimester to prevent prolonged exposure to hypoglycemia in pregnancy.

### 4.3. Isolation of Circulating Extracellular Vesicles

Circulating EVs were isolated from rat plasma samples, as described previously [29,71]. Plasma samples were thawed and centrifuged at 12,000× *g* for 2 min at 4 °C. The resulting supernatant was transferred to fresh tubes, and samples were further centrifuged at 20,000× *g* for 20 min. Following this, supernatants were discarded, and pellets were resuspended in filtered 1× PBS (for nanoparticle tracking analysis) or an Annexin V binding buffer consisting of 10 mM HEPES, 140 mM NaCl, 2.5 mM CaCl_2_, pH 7.4 (for flow cytometry analysis).

### 4.4. Nanoparticle Tracking Analysis

We assessed the size and quantity of circulating EV isolates using ZetaView PMX110 nanoparticle tracking analysis system (Particle Metrix, Meerbusch, Germany), as previously described [71,72,73,74]. Samples were diluted into the working range of the system and 1 mL of sample was loaded into the fluid cell after system calibration with 105 nm and 400 nm polystyrene beads. Particles were visualized by light microscopy and the Brownian motion of individual particles was tracked by video (11 position cameras).

### 4.5. Quantification of Endothelial EVs by Flow Cytometry

EV isolates were incubated with Annexin V-FITC (1:50, Biolegend) and anti-CD31-PE (1:100, BD Biosciences) for 2 h and then re-centrifuged at 20,000× *g* for 20 min. Labelled pellets were re-suspended in Annexin binding buffer and transferred to flow cytometry tubes for analysis. Isolated EVs were quantified using a Beckman Coulter CytoFLEX S equipped with CyExpert version 2.3.0.84 (Beckman Coulter Inc., Indianapolis, India). A size gate was prepared using ApogeeMix beads (Apogee Flow Systems, Hertfordshire, UK), as previously described. Samples were analyzed using FlowJo™ version 7.6.5 (FlowJo, LLC, Ashland, OR, USA). EVs were defined as events that were approximately 100–1000 nm in size that exhibited greater Annexin V fluorescence than their negative controls. Results are shown as number of (i) annexin V+ (total), or (ii) annexin V+, CD144+ (endothelial) EVs/mL plasma.

### 4.6. Assessment of mtDNA in Circulating Extracellular Vesicles

mtDNA levels in circulating extracellular vesicles were measured using the methods of Soltesz and Nagy with modification [75]. Isolated EV samples were treated with 2 U of DNA I to degrade DNA material on the outside of EVs. Each sample was then equalized to 200 µL with PBS and lysed through the addition of 20 µL Proteinase k included in the QIAamp DNA Mini kit. DNA was then isolated following the Qiagen’s DNA Mini kit protocol with a 2 µL of RNase A stock solution (10 mg/mL) added to the samples before adding buffer AL. DNA was then eluted in 50 µL AE buffer.

For qPCR analysis, each sample was run in duplicate using mitochondrial gene-specific primers for Nd2 (F-ccaaggaattcccctacaca, R-gaaattgcgagaatggtggt) and Cox2 (F-agacgccacatcacctatca, R-cttgggcgtctattgtgctt), respectively. qPCR was performed using PowerUp™ SYBR™ Green Master Mix (Applied Biosystems, A25742) and 3 µL of DNA per reaction for a total of 20 µL reactions. A 7300 Real-Time PCR System (Applied Biosystems) was used to run the samples as follows: 2 min at 50 °C, 2 min at 95 °C followed by 40 cycles of 15 s at 95 °C, and 1 min at 60 °C, and the dissociation step (15 s at 95 °C, 30 s at 60 °C, and 15 s at 95 °C). The relative level of each mtDNA gene was calculated using the formula 2 − (Ct-X), that is a derivation of the 2^−ΔΔCt^ method, and normalized on the mean of the Ct for each primer set.

### 4.7. vWF Measurement

As an additional measure of endothelial damage, we quantified levels of circulating von Willebrand factor using a commercially available enzyme-linked immunosorbent assay (ELISA, Elabscience) according to manufacturer’s instructions. The interassay coefficient of variation was <10%.

### 4.8. NAD Measurement

Total NAD and NADH levels were measured in homogenized aortas as per manufacturer’s instructions. The total pool of NAD (NADX) was extracted, and half of the sample was heated to 60 °C for 30 min to decompose NAD+ while keeping NADH intact. Both NADX and NADH (alone) samples were mixed with NAD cycling enzyme, and absorbance was measured at 450 nm. NADX and NADH were quantified by comparing with NADH standard curve and normalising to mg of protein. The ratio of NAD+ (NADX—NADH) to NADH was calculated.

### 4.9. Echocardiography Assessment in Pregnancy

All echocardiography was performed using the Vevo 2100 high-frequency ultrasound imaging system (Fujifilm-VisualSonics, Toronto, ON, Canada) located in the Small Animals and Materials Imaging Core Facility within the University of Manitoba Central Animal Imaging and Transgenic Core Facility. A technician blinded to the experimental groups captured the echocardiograms and images. Blinded images were analyzed using the Vevo LAB Software Cardiac Measurements Package (Fujifilm-VisualSonics, Toronto, ON, Canada) to assess cardiac structure and function. Rats were imaged under mild anesthesia (sedated with 5% isoflurane and 1.0 L/min oxygen and maintained at 2–3% isoflurane and 1.0 L/min oxygen) during echocardiography. Each rat was placed on a heated ECG platform to maintain body temperature at 37 °C. Hair from the rat chest was removed with clippers and Nair™ (3 in 1 Hair Remover Lotion for Sensitive Skin) prior to echocardiography.

Transthoracic ultrasound was used to assess cardiac morphology and function using brightness (B) mode, motion (M) mode and Doppler imaging. B- and M-mode images from the parasternal long axis and short axis view were used to evaluate left ventricle (LV) internal diameters (chamber sizes; LVID), LV wall thickness (anterior and posterior wall; LVAW and LVPW respectively), interventricular septal thickness (IVS), ejection fraction (EF), fractional shortening (FS), end-diastolic volume, end-systolic volume (LV chamber volumes; EDV and ESV, respectively), and heart rate (HR). Diastolic function was evaluated from two-dimensional ultrasound in combination with tissue Doppler and pulse-wave Doppler from apical long-axis views. With pulse-wave Doppler waveforms, transmitral early (E) and atrial (A) wave peak velocities were obtained to calculate the E/A ratio and isovolumic relaxation time (IVRT). With tissue Doppler mode, peak early (E’) annular velocities and early diastolic myocardial relaxation velocity were obtained to calculate the E/E’ ratio. At minimum, parameters were averaged over four cardiac cycles. After echocardiography was completed, rats regained consciousness in an empty cage.

Female rats underwent in vivo echocardiography prior to breeding (pre-breeding echocardiography) for later comparison to cardiac structure and function in pregnancy. Maternal in vivo echocardiography was later performed at e18.5 to assess heart structure and function. The MS250 (13–24 MHz) transducer was used to obtain female echocardiograms.

### 4.10. Statistical Analysis

Data are presented as the mean +/− standard error of the mean (SEM). GraphPad Prism 7 Software (La Jolla, CA, USA) was used for statistical analyses. *p* < 0.05 was regarded as statistically significant for all analyses.

## 5. Conclusions

In this study we observed maternal vascular and cardiac injury in a rodent model of GDM and show the first evidence of impaired vascular NAD+/mitochondrial signaling. These observations further support the presence of subclinical vascular and cardiac injury in GDM, highlight potential therapeutic targets and suggest a role for EVs as a non-invasive tool for assessing endothelial dysfunction and mitochondrial injury in GDM.

## Figures and Tables

**Figure 1 ijms-23-04970-f001:**
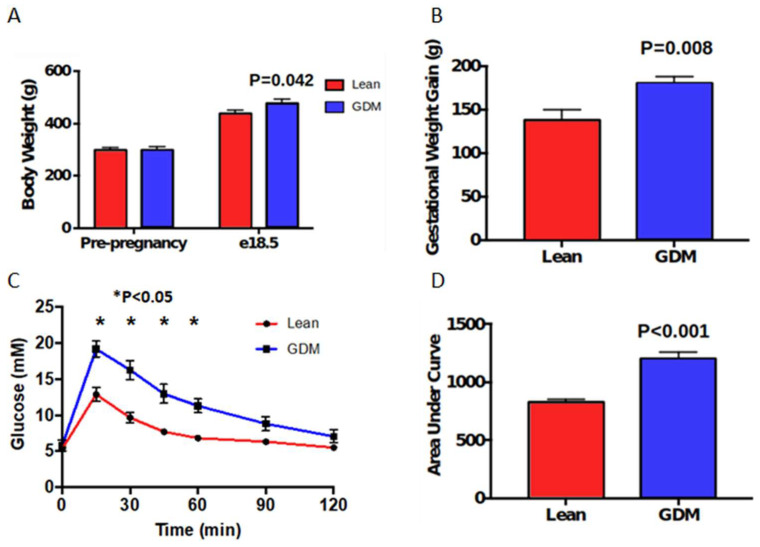
Maternal body composition and metabolic parameters in a rodent model of gestational diabetes mellitus (GDM). (**A**) Pre-pregnancy and late gestational (e18.5) body weight. (**B**) Gestational weight gain (**C**) Late gestational glucose tolerance test (GTT). (**D**) Area under the curve for GTT. n = 10.

**Figure 2 ijms-23-04970-f002:**
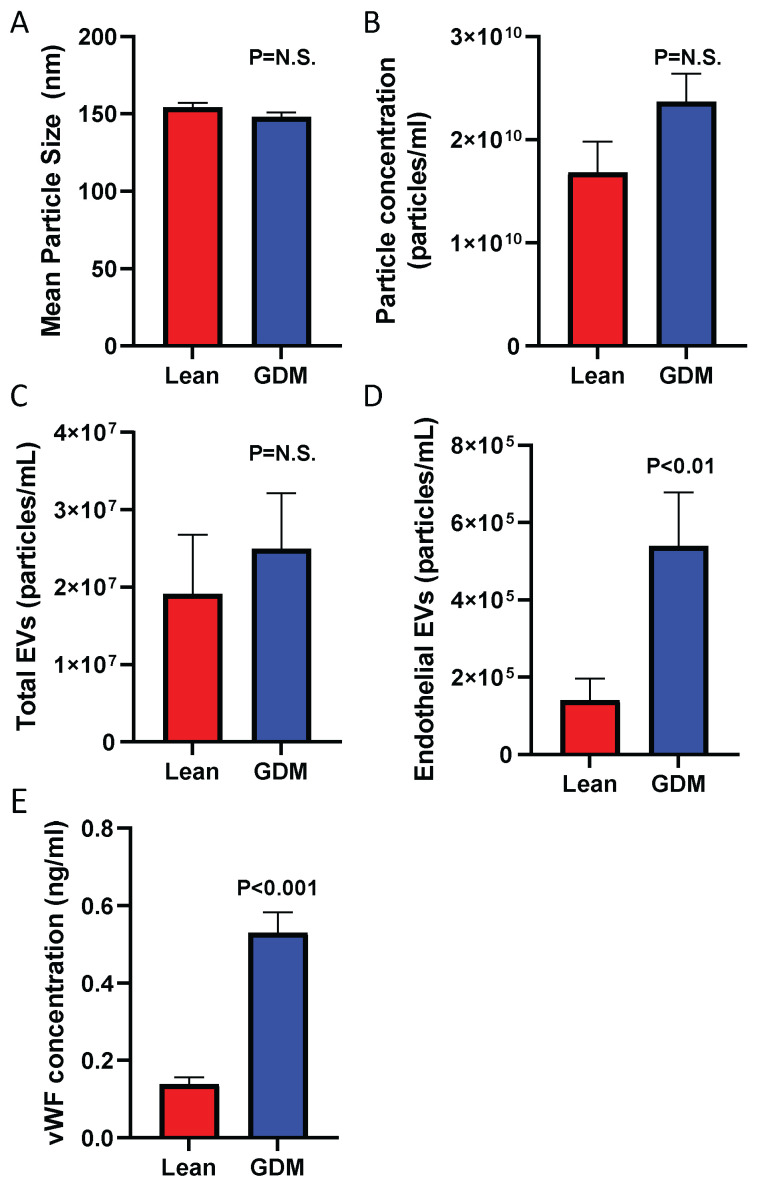
Circulating extracellular vesicles and von Willebrand factor (vWF) in lean and GDM rats. Shown are mean particle size (**A**) and particle concentration (**B**) as assessed by nanoparticle tracking analysis and levels of total (**C**) and endothelial (**D**) EV concentrations by flow cytometry. (**E**) Levels of circulating vWF. n = 10.

**Figure 3 ijms-23-04970-f003:**
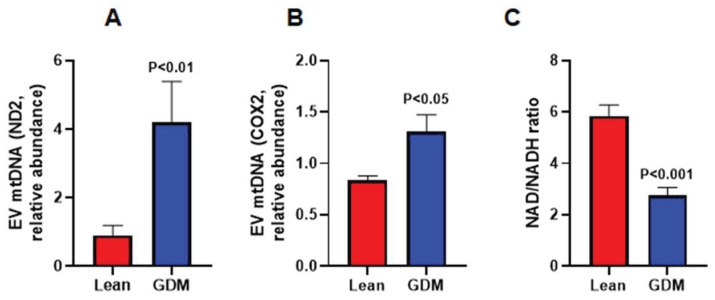
Levels of EV-associated mitochondrial DNA (**A**,**B**) as well as NAD/NADH ratio (**C**) in aortas from lean and GDM rats. n = 6–10.

**Figure 4 ijms-23-04970-f004:**
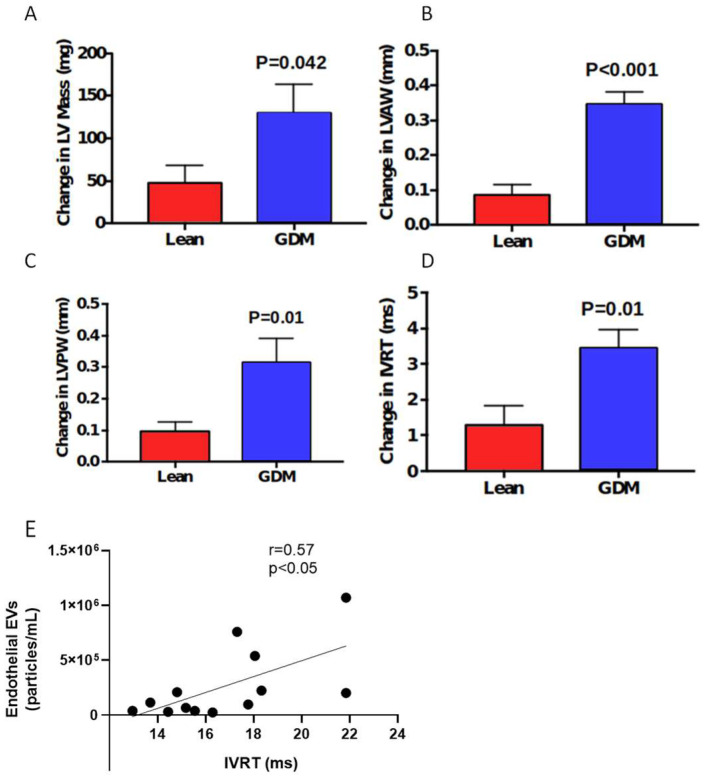
Cardiac injury in a rat model of GDM. Shown are pre-pregnancy to pregnancy changes in left ventricular (LV) mass (**A**), LV anterior wall thickness (**B**), LV posterior wall thickness (**C**), and isovolumetric relaxation time (IVRT **D**). (**E**) Linear regression of association between endothelial EVs and IVRT.

**Table 1 ijms-23-04970-t001:** Echocardiography parameters in female rats.

	Pre-Pregnancy			E-18.5		
	Lean	GDM	*p* Value	Lean	GDM	*p*-Value
LV mass (mg)	662.1 8.0	681.1 11.1	0.70	709.9 9.7	811.3 18.7	0.11
LV mass/g Body Weight	2.21 0.01	2.29 0.03	0.37	1.63 0.02	1.68 0.02	0.52
LVPWd (mm)	1.64 0.02	1.65 0.04	0.93	1.73 0.02	1.96 0.02	0.01 *
LVAWd (mm)	1.52 0.02	1.50 0.03	0.81	1.59 0.02	1.84 0.02	0.01 *
IVSd (mm)	1.42 0.03	1.49 0.02	0.52	1.54 0.03	1.55 0.02	0.95
LVIDd (mm)	7.7 0.07	7.6 0.04	0.81	7.6 0.09	7.2 0.11	0.35
LV Vol (uL)	315.8 6.3	309.0 3.5	0.76	315.3 9.1	280.5 8.9	0.37
EF (%)	74.7 0.9	75.0 1.1	0.94	78.8 0.5	76.8 1.0	0.57
FS (%)	45.6 0.8	46.4 1.2	0.87	49.2 0.5	47.9 1.0	0.69
CO (mL/min)	86.8 1.9	85.6 1.8	0.88	86.7 2.1	80.2 2.5	0.51
SV (uL)	244.2 5.2	238.8 2.8	0.76	240.5 6.4	227.8 6.3	0.64
IVRT (ms)	13.4 0.3	14.7 0.3	0.31	14.6 0.8	18.2 1.0	<0.001 *
E (mm/s)	1021.1 20.8	978.6 24.5	0.68	1251.1 20.8	1104.5 11.3	0.033 *
A (mm/s)	910.0 18.1	821 24.0	0.34	1020.7 23.4	800.9 27.9	0.037 *
E/A	1.21 0.02	1.13 0.02	0.29	1.48 0.04	1.27 0.03	0.16
E’/A’	0.78 0.01	0.83 0.03	0.65	1.19 0.04	1.00 0.01	0.09
E/E’	25.4 0.7	28.9 0.7	0.26	25.4 0.5	30.3 0.8	0.05
MPI	0.35 0.01	0.32 0.01	0.26	0.39 0.01	0.45 0.01	0.04 *
Heart rate (bpm)	355.5 2.8	357.2 5.2	0.92	363.0 3.6	353.1 5.1	0.60

Abbreviations: GDM, gestational diabetes mellitus; LV, left ventricle; d, diastole; LVAW, LV anterior wall; LVID, LV interior diameter; LVPW, LV posterior wall; IVS, interventricular septum; EF, ejection fraction; FS, fractional shortening; CO, cardiac output; SV, stroke volume; IVRT, isovolumic relaxation time; E/A, ratio of mitral valve peak velocities measured by pulsed wave doppler during the early (E) and atrial (A) peaks; E’/A’, ratio of mitral valve peak velocities measured by tissue doppler during the early (E’) and atrial (A’) peaks; E/E’, ratio of early mitral inflow velocity (E) and mitral annular early diastolic velocity (E’) measured by tissue doppler; MPI, myocardial performance index; bpm, beats per minute. * *p* < 0.05 was regarded as statistically significant.

## Data Availability

The data presented in this study are available on request from the corresponding author.

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
