# Peer review of "Extracellular Vesicles as an Index for Endothelial Injury and Cardiac Dysfunction in a Rodent Model of GDM"

_ijms, 2022, doi:10.3390/ijms23094970_

Round 1

Reviewer 1 Report

I realized that the authors did not cite any artcle related relationship between extracellular vesicles and  gestational diabetes mellitus.

The author should cite the appropriate references for relationship between extracellular vesicles and  gestational diabetes mellitus.

There are more than 10 papers for this subject.

Author Response

We would like to thank the reviewer for their constructive feedback which we believe has improved our manuscript significantly. Below please find a point by point response to comments.

Reviewer #1:

  1. I realized that the authors did not cite any artcle related relationship between extracellular vesicles and gestational diabetes mellitus. The author should cite the appropriate references for relationship between extracellular vesicles and gestational diabetes mellitus. There are more than 10 papers for this subject.

Response:

 We agree with the reviewer that the previous version of our manuscript did not adequately address other work regarding the relationship between extracellular vesicles and gestational diabetes mellitus. In our revised manuscript we now include a review of seminal work in this area (see lines 89-98).

Reviewer 2 Report

This article provides numerous interesting insights to direct the investigation on humans and aim to identify the link between endothelial alterations generated by cardiovascular stress factors in pregnancy and possible implications in the pathophysiology of long-term heart damage, in the context of an increasingly important gender approach to cardiology. Through these investigations, an estimate of the risk for the pregnancy itself would be provided as well as an impact on the future of both the mother and the child. Some aspects should be improved:

1) Gestational diabetes represents a risk factor for the development of diabetes in the mother, and diabetes represents one of the main factors promoting heart failure with preserved ejection fraction. In particular, having traced the link with heart failure with preserved ejection fraction is fundamental: this is an increasing percentage compared to all new incident cases of heart failure, and to date there is a lack of ad hoc therapeutic weapons. Please discuss this important aspect and possible clinical perspectives regarding HFpEF  (see Int J Mol Sci. 2021 Jul 17;22(14):7650. doi: 10.3390/ijms22147650.

2) It is very interesting to have traced in the vesicles released by the endothelial cells some real "messages in a bottle" through which a local damage becomes a systemic voice of disease, and can be traced through an extremely limited invasive approach that would make the investigation accessible in terms of primary prevention in the stratification of cardiovascular risk on a large scale. Please briefly discuss the role of new biomarkers in cardiovascular diseases (see Metabolites. 2022 Jan 24;12(2):108. doi: 10.3390/metabo12020108).

4) Fine English editing is required

5) Define all the abbreviations first time they appear in the text, abstract, tables and figures

6) The major limitation is represented by the differences at the anatomical and biochemical level between the mouse model and female gestation, but this work can open up to a future study on the human model. Please briefly discuss future perspectives.

Author Response

We would like to thank the reviewer for their constructive feedback which we believe has improved our manuscript significantly. Below please find a point by point response to comments.

1) Gestational diabetes represents a risk factor for the development of diabetes in the mother, and diabetes represents one of the main factors promoting heart failure with preserved ejection fraction. In particular, having traced the link with heart failure with preserved ejection fraction is fundamental: this is an increasing percentage compared to all new incident cases of heart failure, and to date there is a lack of ad hoc therapeutic weapons. Please discuss this important aspect and possible clinical perspectives regarding HFpEF  (see Int J Mol Sci. 2021 Jul 17;22(14):7650. doi: 10.3390/ijms22147650.

Response:

We agree with the reviewer that HFpEF is a critical consideration given the prevalence in women, lack of biomarkers, therapeutic tools, and high mortality. In our revised manuscript we have expanded our discussion to address HFpEF with specific reference to the suggested manuscript (see lines 228-228).

2) It is very interesting to have traced in the vesicles released by the endothelial cells some real "messages in a bottle" through which a local damage becomes a systemic voice of disease, and can be traced through an extremely limited invasive approach that would make the investigation accessible in terms of primary prevention in the stratification of cardiovascular risk on a large scale. Please briefly discuss the role of new biomarkers in cardiovascular diseases (see Metabolites. 2022 Jan 24;12(2):108. doi: 10.3390/metabo12020108).

Response:

We agree that the previous version of the manuscript did not adequately stress the value of endothelial EVs as biomarkers, nor did it adequately highlight the importance of biomarkers as tools in the management of diabetes. We have revised the discussion to address this important point and now include reference to the suggested manuscript. (see lines 242-252).

4) Fine English editing is required

Response:

We have reviewed and corrected the language in the revised manuscript

5) Define all the abbreviations first time they appear in the text, abstract, tables and figures

Response:

We have revised the manuscript and abbreviations are now defined at first use throughout the manuscript.

6) The major limitation is represented by the differences at the anatomical and biochemical level between the mouse model and female gestation, but this work can open up to a future study on the human model. Please briefly discuss future perspectives.

Response:

We agree that the previous manuscript did not fully acknowledge the differences in rat and human gestation. We have now expanded discussion on this in our study limitations (see lines 255-262).

Round 2

Reviewer 1 Report

Extracellular vesicles as an index for endothelial injury and cardiac dysfunction in a rodent model of GDM -review

Already a couple of paper regarding GDM and Extracellular vesicles as follows.

doi: 10.1016/j.placenta.2021.02.012

doi: 10.1111/aji.13361

doi: 10.3390/biomedicines10020462

doi: 10.1210/jc.2018-02693

DOI: 10.1080/20013078.2019.1617000

doi: 10.1371/journal.pone.0218616

doi: 10.1080/20013078.2019.1625677

This paper focused on cardiac dysfunction in GDM and Extracellular vesicles, the effects of cardiac dysfunction must be clarified. The authors will need discuss role of cardiac function on GDM and Extracellular vesicles.

Author Response

We would like to thank the reviewer for their constructive feedback. We have revised the manuscript to incorporate the recommended changes. Below please find a point by point response to comments.

  1. Already a couple of paper regarding GDM and Extracellular vesicles as follows.

doi: 10.1016/j.placenta.2021.02.012

doi: 10.1111/aji.13361

doi: 10.3390/biomedicines10020462

doi: 10.1210/jc.2018-02693

DOI: 10.1080/20013078.2019.1617000

doi: 10.1371/journal.pone.0218616

doi: 10.1080/20013078.2019.1625677

Response:

In the revised manuscript we have expanded our introduction to include these key manuscripts and reviews as well as to acknowledge the study of EVs in other biological fluids and the putative causal role of EVs in insulin resistance. (see lines 89-99)

  1. This paper focused on cardiac dysfunction in GDM and Extracellular vesicles, the effects of cardiac dysfunction must be clarified. The authors will need discuss role of cardiac function on GDM and Extracellular vesicles.

Response:

We have revised our discussion to address the relationship between cardiac dysfunction in GDM and EVs (see lines 229-239).

Round 3

Reviewer 1 Report

The authors should compare with the other published results and discuss with those results.

Author Response

We would like to thank the reviewer for their constructive feedback. We have revised the discussion to incorporate the recommended changes (detailed below)

  1. The authors should compare with the other published results and discuss with those results.

Response:

We have revised our discussion extensively to compare our results with other studies that have focused on circulating EVs in GDM (see lines 202-230).